# Characterization of Severity in Zellweger Spectrum Disorder by Clinical Findings: A Scoping Review, Meta-Analysis and Medical Chart Review

**DOI:** 10.3390/cells11121891

**Published:** 2022-06-10

**Authors:** Mousumi Bose, Christine Yergeau, Yasmin D’Souza, David D. Cuthbertson, Melisa J. Lopez, Alyssa K. Smolen, Nancy E. Braverman

**Affiliations:** 1Department of Nutrition and Food Studies, College of Education and Human Services, Montclair State University, Montclair, NJ 07043, USA; lopezm41@montclair.edu (M.J.L.); smolena1@montclair.edu (A.K.S.); 2Department of Human Genetics, McGill University, Montreal, QC H4A 3J1, Canada; yasmin.dsouza@mail.mcgill.ca; 3Health Informatics Institute, College of Medicine, University of South Florida, 3650 Spectrum Blvd., Tampa, FL 33612, USA; david.cuthbertson@epi.usf.edu; 4Departments of Human Genetics and Pediatrics, McGill University, Montreal, QC H4A 3J1, Canada; nancy.braverman@mcgill.ca

**Keywords:** Zellweger spectrum disorder, peroxisome biogenesis disorder, signs and symptoms, *PEX* genes, disease severity, scoping review, medical chart review, seizure disorder, feeding difficulties, renal cysts, survival, hexacosanoic acid

## Abstract

Zellweger spectrum disorder (ZSD) is a rare, debilitating genetic disorder of peroxisome biogenesis that affects multiple organ systems and presents with broad clinical heterogeneity. Although severe, intermediate, and mild forms of ZSD have been described, these designations are often arbitrary, presenting difficulty in understanding individual prognosis and treatment effectiveness. The purpose of this study is to conduct a scoping review and meta-analysis of existing literature and a medical chart review to determine if characterization of clinical findings can predict severity in ZSD. Our PubMed search for articles describing severity, clinical findings, and survival in ZSD resulted in 107 studies (representing 307 patients) that were included in the review and meta-analysis. We also collected and analyzed these same parameters from medical records of 136 ZSD individuals from our natural history study. Common clinical findings that were significantly different across severity categories included seizures, hypotonia, reduced mobility, feeding difficulties, renal cysts, adrenal insufficiency, hearing and vision loss, and a shortened lifespan. Our primary data analysis also revealed significant differences across severity categories in failure to thrive, gastroesophageal reflux, bone fractures, global developmental delay, verbal communication difficulties, and cardiac abnormalities. Univariable multinomial logistic modeling analysis of clinical findings and very long chain fatty acid (VLCFA) hexacosanoic acid (C26:0) levels showed that the number of clinical findings present among seizures, abnormal EEG, renal cysts, and cardiac abnormalities, as well as plasma C26:0 fatty acid levels could differentiate severity categories. We report the largest characterization of clinical findings in relation to overall disease severity in ZSD. This information will be useful in determining appropriate outcomes for specific subjects in clinical trials for ZSD.

## 1. Introduction

Peroxisomes are membrane-bound organelles found within almost all eukaryotic cells. Contained within the peroxisomes of cells are numerous enzymes required for normal lipid metabolism and many other biochemical processes necessary for normal health and development [1]. Inherited peroxisomal disorders in humans are generally either due to single peroxisomal enzyme or protein defects, or disorders of overall peroxisome biogenesis, which result in defective biosynthesis, assembly, and general functionality of peroxisomes. Peroxisome biogenesis disorders (PBDs) are primarily caused by mutations in any of 14 different *PEX* genes, which code for peroxins, proteins involved in peroxisome assembly, importation of peroxisomal matrix protein, peroxisome proliferation, and fission [2]. PBDs are divided into two groups: Zellweger spectrum disorder (ZSD) and rhizomelic chondrodysplasia punctata type 1 [3,4].

Zellweger spectrum disorder (ZSD) is an autosomal recessive disorder with a cumulative incidence of ~1:50,000 births [5], although recent newborn screening initiatives may provide updated estimates [4]. Patients diagnosed with ZSD present with extensive clinical heterogeneity and symptoms across multiple organs as a result of decreased or absent peroxisome function. Clinical manifestations include low muscle tone, facial dysmorphisms, impaired growth, sensory and neurological dysfunction, renal and endocrine insufficiency, skeletal abnormalities, and developmental delays [6,7,8,9,10,11,12,13,14,15,16,17,18,19,20].

ZSD can be diagnosed by demonstrating abnormalities in several peroxisome biochemical functions that can be monitored in bodily fluids and tissues. Specific functions of peroxisomes include β-oxidation of very long chain fatty acids (VLCFA) and pristanic acid, α-oxidation of phytanic acid and other metabolic activities including pipecolic acid metabolism, biosynthesis of bile acids and ether glycerophospholipid (plasmalogen). As such, the consequence of impaired or abolished peroxin protein function on peroxisome function generally affects several of these pathways in which metabolites can be measured for biochemical diagnosis of ZSD [1,21]. Genetic testing is also strongly recommended for patients suspected to have ZSD. Pathogenic variants in the *PEX1* gene account for nearly 2/3 of all PBD-ZSD cases, and over a third of cases are caused by pathogenic variants in any of *PEX6*, *PEX12*, *PEX26*, *PEX10*, *PEX2*, *PEX5*, *PEX13*, *PEX16*, *PEX3*, *PEX19*, *PEX14*, and *PEX11β* (ordered from most frequent to least frequent genetic cause of ZSD) [4]. The consequences of the *PEX* gene mutations on the residual peroxin protein function is generally associated with the extent of peroxisome dysfunction and consequent severity of biochemical and clinical phenotypes. For example, patients with two null *PEX1* mutations generally have more severe peroxisome metabolic profile and clinical phenotype compared to patients with the common PEX1-p.G843D missense allele [22,23,24].

Due to the phenotypic variability across the different presentations, ZSD was originally described as multiple distinct syndromes including Zellweger syndrome (ZS), neonatal adrenoleukodystrophy (NALD), infantile Refsum disease (IRD), and Heimler syndrome [25,26]. Attributing to their shared peroxisomal basis, these conditions are now recognized within the overall classification of ZSD, ranging from severe, intermediate, and milder phenotypes [21]. However, current designations of disease severity are often arbitrary and left to the discretion of the medical care provider for the individual patient at that point in time.

Current treatment options for ZSD are relatively limited. The primary standard of care for ZSD is mainly symptomatic, and varies from patient to patient based on symptoms [21,27]. Given the variability of the ZSD spectrum, specific prognosis for ZSD is not well-characterized, aside from the understanding that ZSD is a life-limiting disorder [16,28,29]. This uncertainty in prognosis and treatment effectiveness leaves family caregivers for patients with considerable stress and fear [30], leaving an impact on the whole family affected by ZSD.

Research for treatment options specific to ZSD have been limited to primarily small, open label, single group or case report studies. Three case studies have suggested overall improvement with orthotopic liver transplantation [31,32], but more information on the impact of liver transplantation in ZSD on survival, long-term outcomes, and quality of life are still needed. Recent intervention studies with cholic acid, a primary bile acid, in patients with ZSD have yielded conflicting results in treatment effectiveness [21,33,34,35]. Earlier studies suggested that treatment with the polyunsaturated fatty acid docosahexaenoic acid (DHA) may be useful in ZSD [36,37]; however, a randomized, double-blind clinical trial found no beneficial effect of DHA treatment [38]. Overall, the variability in findings and long-term outcomes in current clinical trials for ZSD has resulted in a limited ability to interpret data and make clear recommendations for therapies in ZSD. Although this variability in treatment effect may be attributable to several factors, the differences in clinical endpoints used for each study as well as the clinical heterogeneity within and across study populations likely plays a considerable role in the lack of a consistent effect in treatment options for ZSD. Some studies have shown that certain biochemical markers used in diagnosis, such as very long chain fatty acids (VLCFA), are predictors of survival [27,39,40], but there are few studies that clearly show an association between VLCFA levels and the clinical impact of ZSD. Moser et al. reported that VLCFA levels were associated with earlier designations of disease severity (ZS, NALD, and IRD); however, it is not described how these designations were characterized [41]. A recent study suggested that VLCFA levels may be related to the age of onset of symptoms in ZSD, but no direct correlations were reported [40]. Additionally, despite a general relationship between ZSD genotype and clinical phenotype, there is a wide range in severity of symptoms as well as clinical heterogeneity within genotypes [42]. This presents major difficulties in predicting severity or prognosis based on biochemical markers or genotype alone. Moreover, this may preclude the development of well-designed clinical trials, where prognosis is variable and thus specific outcomes may not be measurable in some patients. This clinical heterogeneity highlights the importance of developing a robust method for severity designation in ZSD. Indeed, clinical trials for cholic acid therapy and case reports with orthotopic liver transplantation have generally included participants with a milder ZSD phenotype [31,32,33,34,35,42], although criteria for milder ZSD were not clearly defined in these studies.

A recent study described the development of a severity scoring system in ZSD to be used for prognostic measurements and clinical trial stratification. This scoring system used a 3-point severity scale in 14 clinical domains and validated this on 30 ZSD patients [43]. Although the development of this scoring system is a first step in better clinical characterization of ZSD, there were limitations in the utility and discriminatory ability in of some of the categories selected, as well as the relative homogeneity of the patient cohort used to validate the study.

As with many other rare diseases, there is a need for a more comprehensive clinical characterization of ZSD to assist clinical management and help inform future clinical trials. The majority of published clinical reports on ZSD are either case studies or small cohort studies, but collectively, constitute an extensive representation of the broad clinical presentation of ZSD. Moreover, our ongoing natural history study of ZSD [44] is one of the largest clinical studies to date seeking to define the clinical phenotypes, progression, and outcomes of ZSD. Taken together, this available evidence may be useful in thoroughly characterizing the clinical presentation of ZSD as well as potentially determining criteria to define severity in ZSD.

The purpose of this study was to conduct a scoping review and meta-analysis of existing literature, and a medical chart review of medical records from our natural history study to calculate the occurrence and frequency of specific symptoms and clinical findings in ZSD. Specifically, we sought to determine whether or not the presence of specific clinical findings in ZSD could be used to characterize and potentially predict severity in ZSD. In doing this, we hope to gain more insight to the impact of ZSD across the spectrum, as well as perform an in-depth characterization of clinical findings within severity categories in ZSD, based on comprehensive evidence. This information will be useful in the development of more-informed clinical therapeutic trials that could potentially improve the quality of life for patients and families affected by ZSD.

## 2. Materials and Methods

### 2.1. Literature Review

The methodology of Peters et al. [45] was used to conduct the literature search for this scoping review. The population of interest identified for this review includes patients diagnosed with ZSD. The primary study variables include disease severity for ZSD, clinical findings, and survival. Additional variables that were included in the analysis for each study include age of patients (either at the time of study or at death), gender, and type of study (case study, retrospective chart analysis, or clinical cohort). Criteria for inclusion in this review were primary clinical research studies published in English that characterized severity (implicitly or explicitly) in patients diagnosed with ZSD. Explicit characterization of severity included descriptors such as mild, intermediate, or severe when describing subjects. Implicit characterization of severity was generally based on using older terminology where NALD represented intermediate ZSD and IRD/Heimler syndrome represented milder forms of ZSD. For any studies in which determination of severity upon initial review was not clear, preliminary severity criteria (based on clinical findings—see Appendix A) selected as a consensus by our research team were used to assess whether severity could be determined in that study. The selection of the severity criteria was based on known cardinal clinical features in each severity group [21,46] and on information collected in our natural history study. From these criteria, we developed a preliminary severity designation flowchart to evaluate patients. Those with four or more clinical findings among neonatal seizures, hypotonia, failure to thrive, age at death before age 2 years, polymicrogyria on brain MRI, bilateral renal cortical microcysts on ultrasound and chondrodysplasia punctata on hips or knees on X-rays were classified as severe patients, patients presenting with 2 or more clinical findings among adrenal insufficiency, gastroesophageal bleeding, tube feeding, inability to walk independently, and inability to communicate with at least 2–3-word sentences after age 2 years were classified as intermediate patients. Remaining patients were classified as mild.

Studies that only included patients with other peroxisomal disorders were excluded, and studies that did not include clinical findings were excluded. Studies that only described symptoms in patients after onset of an aggressive inflammatory leukodystrophy were excluded, as this type of leukodystrophy is characterized by rapid neurological regression that can occur in both mild and intermediate phenotypes and therefore obscures the natural history of peroxisome dysfunction [47,48,49]. Other exclusion criteria were studies where severity was not assigned by the authors and could not be assigned based on our consensus criteria.

For the purpose of this review, we conducted multiple literature searches in PubMed (https://pubmed.ncbi.nlm.nih.gov/, accessed on 11 May 2022) using the search term combinations of “Zellweger spectrum disorder” AND “Clinical”, “Zellweger syndrome” AND “Clinical”, “Neonatal adrenoleukodystrophy” AND “Clinical”, “Infantile Refsum disease” AND “Clinical”, and “Heimler syndrome” AND “Clinical”. Papers were initially screened by their titles and abstracts only. After the initial screening, full texts of the papers were read and mined for relevant content. Additionally, a manual search of reference lists from various articles was also conducted to identify any additional articles that were suitable for inclusion. Five research team members (MB, CY, YD, MJL, and AKS) participated in the literature search activities to ensure consensus across inclusion of studies.

A synthesis matrix was used for charting data in the studies [50]. The synthesis matrix form was developed by the research summarizing each primary reviewed article by: author, year of publication, type of study, study purpose, age and gender of subjects, degree of severity, clinical findings, and survival. Results were first grouped by type of study (case study vs. population/cohort study and then by clinical findings). Case studies were identified as a study that described 2 or fewer subjects. Population/cohort studies were identified as a study that described 3 or more subjects. Clinical findings were further categorized as whether subjects had: seizure disorders, abnormal EEG, brain abnormalities on MRI or autopsy (white matter abnormalities or structural abnormalities such as neuronal migration defects, subependymal germinolytic cysts, cerebral or cerebellar atrophy, and ventricle dilation), ataxia, hypotonia, mobility difficulties (reduced or no ambulation), verbal communication difficulties (based on delayed, lack of or reduced speech, inability to form sentences, slow speech), hearing loss, vision loss, feeding difficulties (based on orogastric, nasogastric, gastrostomy, or jejunostomy tube feeding, poor sucking reflex, poor or inability to swallow, dysphagia, inability to feed independently, chronic choking or gagging while feeding, or consumption of modified or medical foods), gastroesophageal reflux (GER—based on chronic use of GER medication), abnormal liver function or structure (based on the presence of hepatomegaly, palpable liver, liver fibrosis, cholestasis, chronic jaundice or elevated alanine aminotransferase, aspartate aminotransferase, alkaline phosphatase or gamma-glutamyl transferase levels in blood), bilateral renal cortical microcysts, kidney stones/hyperoxaluria/nephrocalcinosis, adrenal insufficiency (based on an explicit diagnosis or chronic use of corticosteroid medication), chronic respiratory symptoms or support (respiratory distress or failure, chronic respiratory infections, need for ventilation, oxygen or other respiratory support), bone fractures, low bone mineral density, dental abnormalities (based on delayed eruption enamel hypoplasia, amelogenesis imperfecta, weak enamel, crowding and/or enamel abnormalities), and shortened lifespan (died ≤ age 2 years). These categories were chosen based on an existing symptom inventory for ZSD developed by multiple stakeholders in ZSD (expert clinicians, family caregivers, and researchers) [10]. Upon further evaluation of the natural history data, additional symptom categories of global developmental delay, intellectual disability, and failure to thrive at birth were included. An “other” field for symptoms was also included in the synthesis matrix to account for any additional symptoms that were not included in the categorization.

### 2.2. Review of Medical Charts from Our Longitudinal Natural History Study on PBD-ZSD

We collected information from medical records from 150 individuals with PBD-ZSD from the United States, Canada, South America, Europe, Asia, Middle East, and Australia enrolled in our IRB-approved Longitudinal Natural History Study on peroxisomal disorders at the McGill University Health Center (MUHC) from January 2012 to January 2021 (Study #11-090-PED, clinicaltrials.gov identifier NCT01668186). Individuals enrolled in this study already received a diagnosis of ZSD based on molecular and biochemical testing at their local health care centers and were included in this study after obtaining informed consent. We obtained medical records from each participant’s local health care institution after we received the authorization from the patient or the parent/legal representative. We reviewed the medical records and input relevant medical data into our custom-made Microsoft Access database.

We extracted clinical data for each from our database and reported in a data synthesis matrix whether the individual had presence or absence of the following symptoms: aggressive demyelinating leukodystrophy (defined as reported brain MRI showing extensive gadolinium enhancement or demyelination and loss of at least one of gross motor, fine motor, communication, and eating and drinking abilities in the period from 12 months prior to the brain MRI to 12 months after the brain MRI), based on the documented natural history of ccALD [47,48,49], seizure disorder (reported by neurologist or chronic use of anti-epileptic medication), abnormal EEG, brain MRI abnormalities (as defined above), ataxia, hypotonia (from a neurology examination), global developmental delay (reported by any healthcare professional or if delays reported in gross motor, fine motor, and cognitive abilities), mobility at best point (not sitting independently, sitting independently or crawling, walking with support or walking independently, reported by any healthcare professional), verbal communication (no words, less than 50 words, 2 or 3 words together or full sentences, reported by any healthcare professional), intellectual disability (reported by any healthcare professional), hearing loss (from audiology evaluations), vision loss (from ophthalmology evaluations), feeding difficulties (nasogastric, gastrostomy or gastro-jejunal tube feeding, exclusively or not), failure to thrive (diagnosed by any healthcare professional), GER (as defined above), abnormal liver functions in blood (as described above), abnormal liver structure (hepatomegaly reported by any healthcare professional and/or structural abnormalities on ultrasound, including coarse liver, heterogeneous appearance, echogenic liver, fibrotic liver, nodules/cancer, portal hypertension), bilateral renal cortical microcysts (from renal ultrasound), kidney stones/nephrocalcinosis (from renal ultrasound or X-rays), adrenal insufficiency (as defined above), chronic respiratory symptoms (chronic use of asthma medication and/or respiratory support), cardiac abnormalities (abnormal findings on electrocardiogram, echocardiogram, or on chest X-rays), bone fractures (from X-rays), low bone mineral density (from dual-energy X-ray absorptiometry scan), dental abnormalities (as defined above, reported by a dentist), and shortened lifespan (as defined above). Most clinical findings that were evaluated in the natural history data were evaluated similarly in the literature review with the exception of communication and mobility, which were evaluated with respect to more specific outcomes in the natural history study due to the more uniform availability of data in subjects.

To evaluate prevalence of selected clinical findings across disease severities in patients in our ZSD natural history study, we separated our patients into three phenotypic severity subgroups (severe, intermediate, and mild). We used our collected clinical data to separate patients into these three subgroups by following our severity designation flowchart as described above. Genotypes were available for most patients but were not taken into consideration for evaluating phenotypic severity. We also reported gender, current age, age at death, and levels of peroxisome markers in blood, including plasma hexacosanoic acid (C26:0) very long chain fatty acid, pristanic acid, dihydroxycholestanoic acid (DHCA), trihydroxycholestanoic acid (THCA), C16:0 DMA/C16:0, and C18:0 DMA/C18:0 plasmalogens. For all data listed above, we also documented the first age at which they were reported in the medical records. The availability of specific information on these findings varied for each patient depending on whether it was reported in the medical charts obtained.

We calculated the prevalence of each clinical finding as the number of patients with the symptom out of the number of patients with available assessments and the associated means and standard deviations of first age reported by symptom, for all patients as well as for each disease severity.

### 2.3. Statistical Analyses

For the cohort studies, case studies and natural history study analyses, the percentages of subjects within each severity category who presented with each clinical finding were calculated and presented as either percentages with associated prevalence proportion (meta-analyses of cohort and case studies) or percentages with total number (*n*) of subjects evaluating for each outcome variable (natural history study). For the cohort data, these percentages were weighted. Ages of onset for the clinical findings as well as levels of peroxisome markers from the natural history study were presented as medians with associated first and third quartiles and compared across the severity categories with either the Wilcoxon Rank Sum Test when only pairwise analyses were possible, or with the Kruskal–Wallis Test when data was available for all three severity categories. Chi-square tests or Fisher’s Exact tests were used to determine whether the clinical finding could, overall, distinguish between the three severity categories (mild, intermediate, and severe). Similarly, pairwise comparisons were performed between the mild and intermediate categories, and between the intermediate and severe categories. Due to multiple testing, a *p*-value of 0.025 was considered significant for these pairwise comparisons. Kaplan–Meier curves and the log rank test were used to estimate and test for differences in survival times between severity groups for each dataset. Survival probability was presented as a percentage ± standard error. Using the natural history study data, the number of significant univariable clinical findings was calculated both for subjects who had ever presented with the clinical finding and presented before the age of 2 years. Univariable multinomial logistic models were used to determine if these parameters, as well as plasma C26:0 fatty acid levels, could distinguish between the severity categories. Multivariable multinomial logistic models were considered, but not presented due to multicollinearity. Severity category probability was presented as a percentage with the associated 95% CI. Other than those noted, all analyses with *p*-values < 0.05 were considered significant. Statistical analyses were performed using SAS v9.4 (Cary, NC, USA).

## 3. Results

### 3.1. Literature Search Results

A primary search on PubMed using the identified keywords yielded 468 results. After excluding papers based on title and abstract, 126 full text articles were evaluated from the literature search. Thirty-two additional articles were identified from our manual literature search from reference lists; 11 were duplicates and therefore removed. Thirty-eight of the articles were population/cohort studies (*n* > 2 subjects), while 109 studies were case studies (*n* = 1–2 subjects), for a total of 147 full-text articles that were reviewed for potential inclusion in the literature review. Sixteen of the population/cohort studies and 24 of the case studies were excluded due to either a lack of severity designation, the fact that symptoms were grouped across mixed severity levels, there was considerable redundancy in subjects across studies, or that patients’ symptoms were described after the onset of an aggressive demyelinating leukodystrophy. Ultimately, we reviewed 22 population/cohort studies and 85 case studies, for a total of 107 articles that were included in the review (Figure 1).

### 3.2. Severity Designation in Cohort and Case Studies

Of the 22 population/cohort studies, 5 studies were identified as describing a cohort of severe ZSD patients, 5 studies were identified as describing intermediate ZSD patients, and 7 studies were identified as describing mild ZSD patients. Two studies described patients in both the severe and intermediate category, and 1 study described patients in both the intermediate and mild category. Two studies described patients in all three severity categorizations.

Of the 85 case studies, 36 studied patients in the severe category, 16 studied patients in the intermediate category and 31 reported on patients in the mild category of severity. Two case studies included one severe and one intermediate patient. For both cohort and case studies, most did not evaluate all symptoms and clinical findings that were included as outcomes for this review.

### 3.3. Severity Designation in Natural History Study

We collected data from our ZSD natural history database for 150 patients. These patients were composed of both males (52.7%) and females (47.3%) mostly from North America and with varying types mutations in any of the nine *PEX* genes, with the most common alleles being *PEX1* c.2528G > A (p.Gly843Asp) and *PEX1* c.2097insT (p.I700Yfs42X) (more detailed demographics on the patients in the natural history study are presented in Appendix A). Using our preliminary severity designation flowchart (Appendix A) there were 23 patients that fit the criteria for the severe designation, 64 patients in the intermediate designation, 49 patients in the mild designation, and 14 patients whose severity could not be determined due to lack of data, for a total of 136 patients kept for the symptom prevalence analysis.

### 3.4. Characterization of Severe Patients

#### 3.4.1. Cohort and Case Study Meta-Analysis

There were 9 cohort studies (Table 1) [12,51,52,53,54,55,56,57,58] and 38 case studies (Table 2) that described 119 patients (72 from cohort studies, 46 from case studies) with severe ZSD [59,60,61,62,63,64,65,66,67,68,69,70,71,72,73,74,75,76,77,78,79,80,81,82,83,84,85,86,87,88,89,90,91,92,93,94,95,96], although most studies did not evaluate all the symptoms and other clinical findings that were included in this review. The prevalence of seizures (70.8%, out of *n* = 72 total) and brain abnormalities (69.4%, out of *n* = 49 total) in the cohort studies was significantly greater in patients in the severe category compared to those in the intermediate category (*p* ≤ 0.025, Table 1), but these differences between groups were not observed in the case studies (Table 2). The prevalence of renal cortical microcysts in the cohort studies was significantly higher in severe patients (41.9%, out of *n* = 43 total, Table 1) compared to intermediate patients (*p* < 0.001); similar findings were observed with the case studies (Table 2). Interestingly, the prevalence of adrenal insufficiency (7.1%, out of *n* = 14 total), hearing loss (36.8%, out of *n* = 19 total), and vision loss (58.5%, out of *n* = 53 total) in the cohort studies was significantly lower in severely affected patients compared to patients in the intermediate category (*p* ≤ 0.002), but this difference was not found in the cohort studies (Table 1). There were significantly more patients in the severe category that passed away before age 2 observed in both the cohort (83.3%, out of *n* = 54 total) and case (93.0%, out of *n* = 43 total) studies compared to patients in the intermediate category (*p* < 0.001, Table 1 and Table 2).

#### 3.4.2. Natural History Study

We collected clinical findings from 23 patients with severe ZSD. The prevalence of seizures (100%, out of *n* = 23 total), abnormal EEG results (100%, out of *n* = 17 total), bilateral cortical microcysts (79%, out of *n* = 19 total), and cardiac abnormalities (81.3%, out of *n* = 16) was significantly greater in patients in the severe category compared to those in the intermediate category (*p* ≤ 0.025, Table 3). Similar to our meta-analysis, the prevalence of adrenal insufficiency (14.3%, out of *n* = 21 total) was significantly lower in severely affected patients compared to patients in the intermediate category (*p* = 0.002). Ages of onset for these clinical findings were significantly lower in severe patients compared to intermediate patients (median ages range from 0 to 0.1 years and from 0.1 to 3.8 years, respectively). There were significantly more patients in the severe category that passed away before age 2 years (95.7%, out of *n* = 23 total) compared to patients in the intermediate category (*p* < 0.001, Table 3). Levels of C26:0 fatty acid were significantly higher in severe patients compared to intermediate patients (medians 3.9 and 2.0 μg/mL, respectively, *p* < 0.001, Table 4). Similarly, levels of THCA in blood were significantly higher in severe patients compared to intermediate patients (medians 22.5 and 0.6 μmol/L, respectively, *p* < 0.05, Table 4). Levels of erythrocyte C16:0 DMA/C16:0 and C18:0 DMA/C18:0 plasmalogen ratios were significantly lower in severe patients compared to intermediate patients (*p* < 0.001, Table 4).

### 3.5. Characterization of Intermediate Patients

#### 3.5.1. Cohort and Case Study Meta-Analysis

There were 10 cohort studies [12,28,29,39,51,58,98,99,100,101] and 18 case studies [60,93,101,102,103,104,105,106,107,108,109,110,111,112,113,114,115,116] that described 77 (54 from cohort studies, 23 from case studies) patients with an intermediate form of ZSD. In the cohort studies, the prevalence of seizures (51.0%, out of *n* = 51), brain abnormalities (15.4%, out of *n* = 26), hypotonia (97.6%, out of *n* = 41), reduced mobility (70.8%, out of *n* = 24), feeding difficulties (52.9%, out of *n* = 17), abnormal liver function or structure (79.0%, out of *n* = 38), adrenal insufficiency (57.1%, out of *n* = 28), and hearing loss (57.1%, out of *n* = 28) was significantly greater in the intermediate category compared to the mild category (*p* ≤ 0.011, Table 1). There were no significant differences in symptoms or clinical findings between the intermediate and mild categories in the case studies. There were significantly more patients in the intermediate category that passed away before age 2 observed in both the cohort (24.4%, out of *n* = 45 total) and case (31.3%, out of *n* = 16 total) studies compared to patients in the mild category (*p* ≤ 0.025, Table 1 and Table 2).

#### 3.5.2. Natural History Study

We collected clinical findings from 64 patients with intermediate ZSD. The prevalence of seizures (41.3%, out of *n* = 63 total), hypotonia (72.2%, out of *n* = 36 total), global developmental delay (97.5%, out of *n* = 40 total), not sitting independently and walking with support as best mobility milestones reached (28.8%, out of *n* = 59 total and 33.9%, out of *n* = 59 total, respectively), no words as best communication ability reached (71.7%, out of *n* = 53 total), intellectual disability (100%, out of *n* = 23 total), vision loss (89.1%, out of *n* = 55 total), feeding difficulties (71.9%, out of *n* = 57 total), failure to thrive (94.1%, out of *n* = 17 total), gastroesophageal reflux (51.7%, out of *n* = 58 total), abnormal liver functions (92.9%, out of *n* = 56 total), abnormal liver structure (61.4%, out of *n* = 57 total), adrenal insufficiency (54.2%, out of *n* = 59 total) was significantly greater in patients in the intermediate category compared to those in the mild category (*p* ≤ 0.025, Table 3). Walking independently as best mobility milestones reached (23.7%, out of *n* = 59 total), putting 2–3 words together and talking in full sentences as best communication abilities reached (5.7%, out of *n* = 53 total and 1.9%, out of *n* = 53 total, respectively) were significantly less prevalent in intermediate patients compared to patients in the mild category (*p* ≤ 0.005, Table 3). Age of onset for these clinical findings was significantly lower in intermediate patients compared to mild patients (median ages range from 0.2 to 3.8 years and from 0.5 to 20.7 years, respectively), except for the communication and mobility abilities mentioned above, where no significant differences were found in the age of onset. Levels of C26:0 fatty acid were significantly higher in intermediate patients compared to mild patients (medians 2.0 and 0.6 μg/mL, respectively, *p* < 0.001, Table 4). Similarly, levels of DHCA and THCA in blood were significantly higher in intermediate patients compared to mild patients (*p* < 0.05, Table 4). The prevalence of decreased levels of erythrocyte C16:0 DMA/C16:0 and C18:0 DMA/C18:0 plasmalogen ratios (83.3%, out of *n* = 24 total and 91.7%, out of *n* = 24, respectively) was significantly greater in patients in the intermediate category compared to those in the mild category (*p* ≤ 0.025, Table 3). Levels of erythrocyte C16:0 DMA/C16:0 and C18:0 DMA/C18:0 plasmalogen ratios were significantly lower in intermediate patients compared to mild patients (*p* < 0.001, Table 4).

### 3.6. Characterization of Mild Patients

#### 3.6.1. Cohort and Case Study Meta-Analysis

There were 10 cohort studies [8,16,29,42,51,99,117,118,119,120] and 31 case studies [6,11,15,26,32,121,122,123,124,125,126,127,128,129,130,131,132,133,134,135,136,137,138,139,140,141,142,143,144,145,146] that described 124 patients with mild ZSD (80 from cohort studies, 36 from case studies). In the cohort studies, a seizure disorder was reported in 15.9% of patients (out of 63 total), hypotonia was present in 50.0% of patients (out of 42 patients total), reduced mobility was reported in 35.7% of patients (out of 70 patients total), feeding difficulties were reported in 14.8% of patients (out of 54 patients total), abnormal liver function or structure was reported in 54.3% of patients (out of 70 patients total), adrenal insufficiency was reported in 11.8% of patients (out of 34 patients total), and hearing loss was present in 52.0% of patients (out of 75% patients total) (Table 1). Similar prevalence proportions were reported in the case studies; however, the differences between the severity categories were not significantly different from one another in the case studies (Table 2). There were no patients in either of the cohort studies (Table 1) or the case studies (Table 2) that passed away before age 2.

#### 3.6.2. Natural History Study 

We collected clinical findings from 49 patients with mild ZSD. Among these patients, a seizure disorder was reported in 16.3% of patients (out of 49 total), hypotonia was present in 72.1% of patients (out of 43 patients total), global developmental delay was reported in 33.3% of patients (out of 36 patients total), sitting independently was reported in all mild patients (out of 49 patients total), walking with support and walking independently were reported as best mobility abilities in 12.2% and in 87.8% of patients, respectively (out of 49 patients total), ability to communicate with words was found in all patients, with 26.1% of them speaking 2–3 words together and 71.7% of them speaking in full sentences (out of 53 patients total). Intellectual disability was reported in 30% of patients (out of 30 total), vision loss was reported in 67.4% of patients (out of 46 total), none of the patients had feeding difficulties (out of 48 patients total), failure to thrive was reported in 5.3% of patients (out of 19 total), gastroesophageal reflux was reported in 11.1% of patients (out of 45 total), abnormal liver function was reported in 65.9% of patients (out of 44 patients total), abnormal liver structure was reported in 31% of patients (out of 42 patients total), and adrenal insufficiency was reported in 10.4% of patients (out of 48 patients total) (Table 3). Decreased levels of erythrocyte C16:0 DMA/C16:0 and C18:0 DMA/C18:0 plasmalogen ratios were found in 23.5% and in 35.3% of patients, respectively (out of 17 patients total, Table 3). None of the patients in the intermediate and mild categories from our natural history study passed away before age 2 years (out of *n* = 64 and *n* = 49 total, respectively, Table 3).

### 3.7. Other Clinical Findings

The prevalence of all clinical findings and symptoms in ZSD as determined in the cohort, case, and natural history studies are summarized in Table 1, Table 2, and Table 3, respectively. In the cohort studies, there were no significant differences in reports of an abnormal EEG, ataxia, gastroesophageal reflux, kidney stones/hyperoxaluria, low bone mineral density, bone fractures, dental abnormalities, global developmental delay, intellectual disability, verbal communication difficulties, and chronic respiratory symptoms (Table 1). Similar results were found in the case studies (Table 2). In the natural history study, no significant differences were found in ataxia, kidney stones/hyperoxaluria, low bone mineral density, bone fractures, dental abnormalities, and chronic respiratory symptoms. However, prevalence of brain MRI abnormalities and hearing loss was not significantly different across severity groups from the natural history study as opposed to the literature findings. In addition, prevalence of aggressive demyelinating leukodystrophy and elevation of C26:0 fatty acid, pristanic acid, DHCA and THCA levels was not significantly different across severity groups from the natural history study.

### 3.8. Survival of Severe, Intermediate, and Mild Patients

#### 3.8.1. Cohort and Case Studies

Kaplan–Meier survival curves for the cohort and case studies, based on ages available, are shown in Figure 2A,B. In the cohort studies, the probability of survival (% probability ± standard error) for patients in the severe category at age 0–1 years was 36.1 ± 3.4%, whereas for patients in the intermediate and mild categories, the probability of survival at the same age was 75.0 ± 5.3% and 95.8 ± 2.4%, respectively. At age 4–5 years, the probability of survival for patients in the severe category remained at 36.1 ± 26.1%, while the probability of survival at the same age decreased slightly to 59.6 ± 3.6% and 92.8 ± 3.1% for patients in the intermediate and mild categories, respectively. By age 8–9 years, the probability of survival for patients in the severe category decreased sharply to 0%, while for patients in the intermediate and mild categories, the probability of survival at the same age decreased more steadily to 54.6 ± 11.1% and 85.6 ± 4.5%, respectively. In the case studies, the probability of survival for patients in the severe category at age 0–1 years was 26.1 ± 3.3%, whereas for patients in the intermediate and mild categories, the probability of survival at the same age was 81.0 ± 7.9% and 100 ± 0.0%, respectively. At age 4–5 years, the probability of survival for patients in the severe category was 3.7 ± 0.0%, while the probability of survival at the same age was 54.9 ± 21.3% and 92.4 ± 5.3% for patients in the intermediate and mild categories, respectively. By age 8–9 years, the probability of survival for patients in the severe category decreased to 0%, while the probability of survival at the same age remained relatively unchanged for patients in the intermediate and mild categories (54.9 ± 26.1% and 92.4 ± 5.8%, respectively) The log rank test for the survival curves was significant across the three severity categories for both the cohort and the case studies (*p* < 0.001).

#### 3.8.2. Natural History Study

The Kaplan–Meier survival curves for patients from the natural history study are shown in Figure 2C. The probability of survival for patients in the severe category at age 0–1 years was 30.4 ± 5.4%, whereas for patients in the intermediate and mild categories, the probability of survival at the same age was 100%. At age 2–3 years, the probability of survival for patients in the severe category dropped down to 0%, whereas the probability of survival age decreased to 95.3 ± 2.6% for patients in the intermediate category and remained at 100% for patients in the mild category. By age 7–8 years, the probability of survival for patients in the intermediate category decreased to 79.2 ± 6.1%, while it slightly decreased to 97.7 ± 2.3% for patients in the mild category. By age 34–35 years, the probability of survival for patients in the intermediate category continued to steadily decrease to 45.3 ± 33.5%, while it decreased to 81.4 ± 15.7% after being stable for 27 years for patients in the mild category. The oldest patient in our mild patient population passed away at age 44 years while there was one intermediate patient who was 50 years old at the time of our study. The log rank test for the survival curves was significant across the three severity categories (*p* < 0.001).

### 3.9. Predictions of Severity in ZSD

#### 3.9.1. Predicted Probabilities of Disease Severity by Number of Clinical Findings

We used univariable multinomial logistic models on the natural history study data to determine whether the clinical findings analyzed (number of symptoms and levels of plasma C26:0 fatty acid) could distinguish between severe, intermediate, and mild severity categories. Clinical findings from the literature review could not be used for this model as they did not differentiate sufficiently between severity categories. We found that the differences in prevalence of seizure disorder, abnormal EEG, bilateral renal cortical microcysts, and cardiac abnormalities across the three severity categories were robust enough to allow for reliable modeling predictions of severity designation (Figure 3). Patients presenting with four of these findings at any age have a predicted probability (with 95% CI) of 90.59% (78.92–100%) of having a severe designation, 8.91% (0.00–20.04%) of having an intermediate designation, and 0.50% (0.00–1.52%) of having a mild designation, with respect to overall disease severity. In contrast, patients presenting with none of these findings at any age have a predicted probability of 0.67% (0.00–1.80%) of having a severe designation, 45.14% (33.82–56.47%) of having an intermediate designation, and 54.19% (42.78–65.60%) of having a mild designation (Figure 3). Running the model exclusively on clinical findings present before the age of 2 years showed similar trends but with higher probabilities of having a severe disease and lower probabilities of having a mild disease with one or more clinical findings (for the four findings present, probabilities of 98.49% (95.17–100%) and 0% of having a severe and mild designation, respectively) (Figure 2). Detailed results from the model are provided in Appendix A.

#### 3.9.2. Predicted Probabilities of Disease Severity by Plasma C26:0 VLCFA Levels

We used the same univariable multinomial logistic model from the natural history study data to determine whether levels of peroxisome biochemical markers could distinguish between severe, intermediate, and mild severity categories. We found that plasma C26:0 VLCFA was the only biochemical marker, among those we analyzed, that allowed for reliable modeling predictions of severity designation (Figure 4). The lower end of the range of C26:0 fatty acid levels in our patients was 0.2 μg/mL and corresponded to a predicted probability of 0.21% (0–0.64%) of having a severe designation, 7.33% (0–15.18%) of having an intermediate designation, and 92.45% (84.4–100%) of having a mild designation, with respect to overall disease severity. In the higher end of the range, C26:0 fatty acid levels of 5.92 μg/mL corresponded to a predicted probability of 55.46% (14.32–96.59%) of having a severe designation, 44.54% (3.41–85.68%) of having an intermediate designation, and 0% of having a mild overall disease severity. Interestingly, the model showed that C26:0 fatty acid levels of 1.08 μg/mL corresponded to an equal chance of having an intermediate or a mild disease, while levels of 5.18 μg/mL corresponded to equal chance of having an intermediate or a severe disease. Detailed results from the model are provided in Appendix A.

## 4. Discussion

In this study, we used a scoping literature review, a subsequent meta-analysis, and a medical chart review of our natural history data to report the prevalence of clinical findings in ZSD across multiple disease severity categories (severe, intermediate, and mild). Collectively, our study represents the characterization of clinical findings from a total of 443 patients with ZSD, the largest report on clinical findings in ZSD to our knowledge. When comparing patients in the severe disease category to those in the intermediate category, we found significant differences in the prevalence of seizure disorders, abnormal EEG, brain abnormalities, renal cortical microcysts, cardiac abnormalities, and a shortened lifespan. We found significant differences between patients in the intermediate and the mild category of disease when comparing the prevalence of seizure disorders, hypotonia, feeding difficulties, gastroesophageal reflux, mobility, global developmental delay, intellectual disability, verbal communication, vision loss, failure to thrive, abnormal liver function and/or structure, adrenal insufficiency, and a shortened lifespan. These results suggest that certain clinical findings may be useful in the designation of severity in ZSD. Although these differences were not always present in all three of the datasets that we analyzed (cohort meta-analysis, case study meta-analysis, and natural history medical chart data), the majority of these findings were evident in the medical chart review of our natural history dataset, which is primary data and thus the most robust dataset. Differences not observed in the natural history data but observed in the meta-analysis included brain abnormalities and hearing loss. However, the ages of onset of these clinical findings (as well as for other clinical findings) were significantly different across severity categories in our natural history data. Additionally, in our scoping literature review, we included brain abnormalities that were observed by MRI and/or during autopsy, whereas the brain abnormalities reported from the natural history study dataset were solely from MRI reports; this may explain why differences in the prevalence of brain abnormalities were not observed in the natural history dataset.

Another interesting finding in both the cohort and the natural history data is the increased prevalence of adrenal insufficiency among patients in the intermediate disease category compared to those in the severe category. The median age of onset for primary adrenal insufficiency has been recently reported as 6.5 years in a large pediatric cohort [147] and was 3.8 years for the intermediate ZSD patients in our natural history study. Considering the high prevalence of patients with a shortened lifespan (having died within 2 years of age) in the severe category, it is likely that many patients in this severity category were too young at time of death for symptoms of adrenal insufficiency to appear.

Multiple reports on characterization and natural history of clinical findings in individuals with ZSD provided important information on the wide spectrum of manifestations in this disorder [14,16,17,18,20]. However, few identify the common clinical findings specific to disease severity categories and selected proxies for disease severity have been variable across studies. For example, Berendse et al. reported the prevalence of clinical findings in 19 adult individuals with ZSD whose severity was designated by communication abilities only. Peripheral neuropathy was more prevalent in their mildest group while leukodystrophy, liver disease, and nephrolithiasis was more prevalent in the more severe group of their cohort [42]. Our findings are similar comparing symptoms in intermediate and mild patients, which highlights that communication abilities is one factor that contributes to predicting intermediate or mild disease severity. However, our results demonstrate that characterization of severity in ZSD implicates several clinical findings that should be considered for more accurate and consistent severity designation. Berendse et al. also reported that presence of adrenal insufficiency did not correlate with severity in 24 ZSD patients by using motor function and communication as proxies for disease severity [42]. In contrast, we showed that motor function, communication, and adrenal insufficiency were all important clinical findings in the characterization of severity between intermediate and mild ZSD individuals. The larger sample size in our datasets may allow for a more comprehensive understanding of the role of these clinical findings in the characterization of disease severity.

The univariable multinomial logistic model analysis of our natural history data revealed that the presence of a seizure disorder, abnormal EEG, renal cortical microcysts, and cardiac abnormalities can estimate the probability of severity category for a patient with ZSD. Specifically, our analysis suggests that the more of these four clinical findings that a patient presents with, the higher probability of having a more severe disease. Increasing disease severity in ZSD is known to be associated with an increasing number of organ systems involved at earlier ages with some disease features being limited to specific severity groups [21,46] and a severity scoring system based on 14 organs typically affected in ZSD was recently reported [43]. This scoring system was developed by expert clinicians in the field based on their knowledge of the disorder and validated in a subset of ZSD patients. Severity scores in this population correlated with specific symptoms as well as scores from another validated survey instrument that measured additional care needs in pediatric patients with neurodevelopment disabilities. Our model supports certain aspects of this ZSD severity scoring system, including both kidney abnormalities and neurological findings as factors in increasing severity. However, the majority of the patients in this study were considered in the mild category of disease, suggesting that this scoring system may have less application in more intermediate and severe forms of the disease. Additionally, the scoring system included clinical findings rarely observed in ZSD, including anorectal malformations, syndactyly, and tumors. Our model is based solely on the prevalence of clinical findings in a large subset of patients with ZSD, with a much broader phenotypic variation than the patients used to validate the previously published scoring system. To our knowledge, this is the first evidence-based model that may serve as an important step in a comprehensive characterization of ZSD severity as determined by clinical findings.

Similarly, our modeling analysis also allowed for estimating the probability of belonging to a severity category based on C26:0 fatty acid levels. The significant differences in C26:0 fatty acid level across the three severity categories (median levels at 12.6, 6.5, and 1.9 times the upper limit of the normal reference range for severe, intermediate, and mild categories, respectively) and the higher number of patients with available C26:0 fatty acid levels in this study allowed to generate the multinomial logistic model. Specifically, our model shows that increasing C26:0 fatty acid levels is associated with a higher probability of having a more severe form of ZSD. Plasma C26:0 fatty acid levels have been identified as a sensitive diagnostic marker for ZSD [27] and strongly associated with survival in ZSD [39,40]. However, the fact that there are ranges of C26:0 fatty acids levels that correspond to equal or close to equal probabilities of belonging to two severity categories highlights the limitation of using this model alone to predict severity.

Related, all three of our datasets show significant differences in survival across severity categories. In our recent publication on caregiver-reported symptoms in ZSD, we found that caregivers of deceased patients were more likely to report their child in the severe category of disease compared to caregivers of living patients [10]. Our current study shows clear evidence of distinct survival patterns across severity categories. Taken together, our study provides preliminary tools to systematically estimate overall disease severity in ZSD, which may be useful in determination of prognosis and appropriate clinical care for patients.

To date, there have been few published clinical trials that have identified therapeutics that are consistently effective in improving symptoms of ZSD. Martinez and colleagues reported that DHA supplementation in ZSD patients improved multiple clinical symptoms [36,148]; however, a randomized clinical trial found no effect of 1 year DHA supplementation on vision and growth [38]. Recently, multiple groups have reported on the effects of cholic acid therapy in ZSD. Similar to DHA, results have been inconsistent in the effectiveness of cholic acid [33,42]. While factors such as length of time for treatment have been identified as potential confounding variables contributing to these inconsistencies, the variability in severity likely also plays a role in the differences in treatment effects. Variability in biochemical parameters was suggested as a contributing factor in the lack of measurable effectiveness of oral betaine supplementation on biochemical function in ZSD patients (Plourde et al., unpublished results). Moreover, in both DHA and cholic acid therapy studies, baseline symptom presentation was discussed as a factor to consider when evaluating the effectiveness of therapy [36,149]. Our findings may be useful to further characterize clinical presentation and severity in ZSD, which may serve as an important consideration in the design of future clinical trials. Although it is not expected that severity category can be used as an outcome variable that can be altered through time or clinical treatment, the results of our studies may have utility in identifying thorough inclusion criteria for clinical trials, or, in the case of studies that include patients of varying severity levels, may help identify feasible and appropriate endpoints for specific groups of participants based on their severity designation.

Our study has several limitations. First and foremost, our designation of severity categories, particularly in natural history study, was based on criteria developed by our team, which could be viewed as arbitrary as previous studies that identify severity in ZSD. Although we recognize the artifact that these designations may introduce in our analyses, given that there is currently no standard for severity designation in ZSD, we attempted to establish a starting point for clinical characterization of severity. We based these criteria on previous literature [21,46] and our expert consensus evaluation of the medical chart data for our natural history study, which, to our knowledge, is currently the largest population of ZSD patients studied in the world. Additionally, when we used these criteria against the designations of severity in the cohort and case study meta-analyses, most of the publications included in this review assigned severity designations that aligned well with our criteria for severity designation. Our study will be useful in establishing more standardized criteria for severity category designation that will be evidence-based rather than based on expert consensus.

Our three datasets did not show complete consistency in the prevalence of clinical findings across the three severity categories. We reported more significant differences across severity categories in the natural history data compared to the cohort and case study meta-analyses. Beyond the differences between the secondary (cohort and case studies review) and primary (medical chart review) analyses, several cohort studies were excluded from our review due to an inability to characterize clinical findings, severity, or other contextual factors on a subject-level basis. Therefore, our review may not present the whole picture of all of the published clinical cohort findings in ZSD. Additionally, our case studies showed very few significant differences across severity categories. Despite having the advantage of being able to study clinical findings on a subject-level basis, case studies are often published to highlight interesting or unusual findings within a population [150], as opposed to reporting on common clinical characterization of that population. Certain clinical findings may have been emphasized in the case studies that we reviewed, while others may have been understated or omitted altogether. Despite analyzing nearly 90 case studies for this review, the actual number of studies included in the statistical comparisons across severity categories for each clinical finding were relatively small compared to the cohort and natural history data, which may explain fewer instances of significant differences across severity categories in the case study data.

We excluded multiple studies where there appeared to be redundancy in patients included in the reviewed cohort studies or the natural history data. However, we cannot rule out that there were some case studies that overlapped with our natural history dataset. Therefore, the total number of subjects that we reported on across all of our three datasets may have been overestimated. While we expect this overlap to be minimal, continued efforts towards a comprehensive, prospective approach to collecting data on clinical and biochemical findings in ZSD patients will decrease the need for a review of case studies to characterize ZSD.

We did not evaluate plasma C26:0 fatty acid levels in the studies included in our meta-analyses; review of existing peer-reviewed publications on plasma C26:0 fatty acid levels may have allowed for additional modeling data analysis. Future evaluation of the literature is needed to further substantiate the predictive capability of plasma C26:0 levels in disease severity. Similarly, genotype–phenotype correlations were not evaluated in this study as genotypes were not consistently available in the literature review.

We excluded clinical findings observed in patients after the onset of an aggressive demyelinating leukodystrophy, which has been identified as a clinical feature in some patients with ZSD [21], due to the impact that such a condition has on clinical presentation. While an aggressive demyelinating leukodystrophy may impact the relevance of our data in some patients, only 5.8% of patients presented with an aggressive demyelinating leukodystrophy in our natural history study (data not shown), suggesting that our findings would have utility in most patients affected by ZSD.

The majority of the data that we collected on clinical findings were binary in nature (presence or absence). We did not collect qualitative descriptions of clinical findings, such as severity of specific symptoms. Due to the fact that evaluation of all of these datasets were retrospective in nature, the standardization of clinical finding descriptions was limited and thus difficult to compare across studies and across different medical charts. Furthermore, our data from both the natural history study as well as the literature review could not distinguish whether or not clinical findings reported were due to the ZSD diagnosis or independent of the disorder. Future studies are required to qualitatively describe clinical findings in relation to ZSD severity, as it may allow to more accurately define severity categories in ZSD.

We collected literature and evaluated our natural history data that reported on patients of all ages with ZSD. As a result, we expect that our findings are applicable to patients of all ages. Stratification of the modeling analyses by age may have provided further information about disease severity probability with respect to specific age groups. A recent study reported that certain biochemical findings in ZSD patients attenuates with age [99]. However, stratification by age in our datasets would likely decrease sample size and prevent detection of significant patterns of probability within age groups. Additional and more complete data on ZSD patients will be necessary to determine if there are differential effects of age on disease severity with respect to clinical and biochemical findings.

There are limitations to applying our modeling data analyses, in their current state, for use in a clinical or research setting. Our data shows that plasma C26:0 fatty acid levels do appear to predict the severity categories with distinct confidence intervals across severity categories. However, our modeling analysis for clinical findings only allowed for incorporating a fraction of the actual clinical findings that we evaluated and shows some overlap between confidence intervals across severity categories. This may present some difficulty in interpreting the true probability of severity category based on the small number of clinical findings observed. The quantity and type of data collected as part of our natural history study varies for each participant depending on the availability of medical charts, and on the examinations and tests selected by the clinician. Therefore, information about all clinical findings collected for this study was not always available for all patients. Nevertheless, our findings are a first step in creating a robust and precise tool that would include more clinical findings to better predict the probability of disease severity. Comprehensive data on clinical findings, based on a standardized prospective assessment approach, will be necessary in future medical examinations and evaluations of patients to develop such a tool. Our study provides guidance on the specific clinical evaluations necessary to create a thorough and exhaustive characterization of severity in ZSD.

## 5. Conclusions

Our study presents a rigorous evidence-based characterization of severity in ZSD, which serves as a foundational step in the development of more robust tools to classify severity in patients. This continued development can be accomplished with comprehensive and standardized data collection by clinicians and researchers, which should include a thorough evaluation of clinical findings, biochemical markers and genotype. Ultimately, tools such as this will allow for better insight in the natural history, clinical care, and identification of targeted treatment options for ZSD.

## Figures and Tables

**Figure 1 cells-11-01891-f001:**
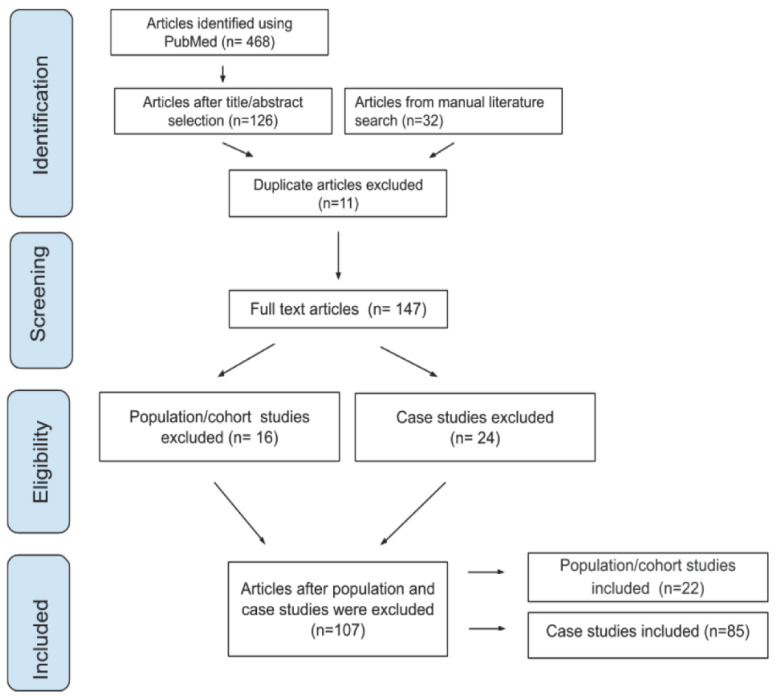
Flowchart showing overview of the literature search and study selection process for the scoping review.

**Figure 2 cells-11-01891-f002:**
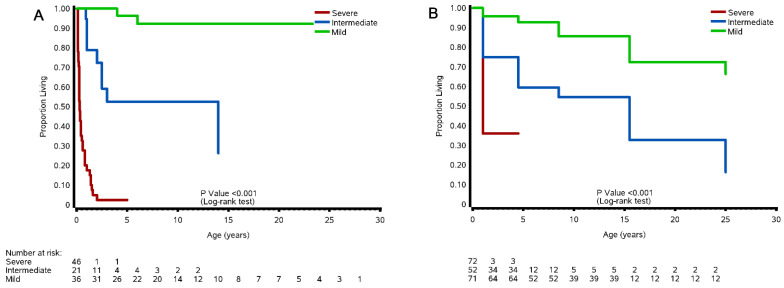
Kaplan–Meier analyses of survival according to severe, mild, and intermediate severity categories in individuals with ZSD. (**A**) Survival for 46 severe (red), 21 intermediate (blue), and 36 mild (green) (total 103) subjects from the case studies is shown; (**B**) survival for 72 severe, 52 intermediate, and 71 mild (total 195) subjects from the cohort studies is shown; (**C**) survival for 23 severe, 64 intermediate, and 49 mild (total 136) subjects from the natural history study is shown. The corresponding log-rank *p* value is shown for each cohort. The numbers at risk (number of surviving patients) are indicated below each curve.

**Figure 3 cells-11-01891-f003:**
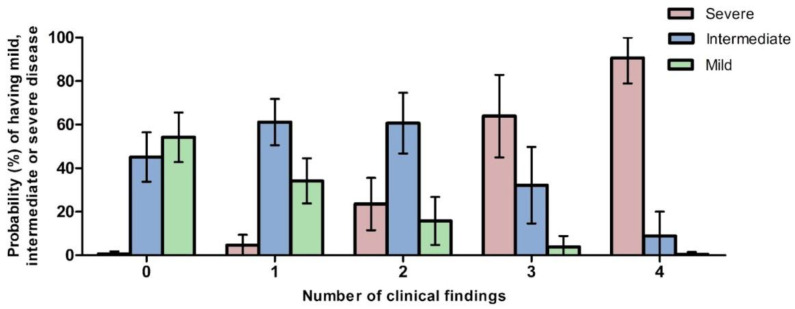
Predicted probabilities of each severity category by number of clinical findings at any age among seizure disorder, abnormal EEG, bilateral renal cortical microcysts, and cardiac abnormalities. Univariable multinomial logistic model was used to predict the probabilities of having severe (red), intermediate (blue), or mild (green) overall disease severity from the number of any of these 4 clinical findings. The 95% confidence intervals are shown as the error bars.

**Figure 4 cells-11-01891-f004:**
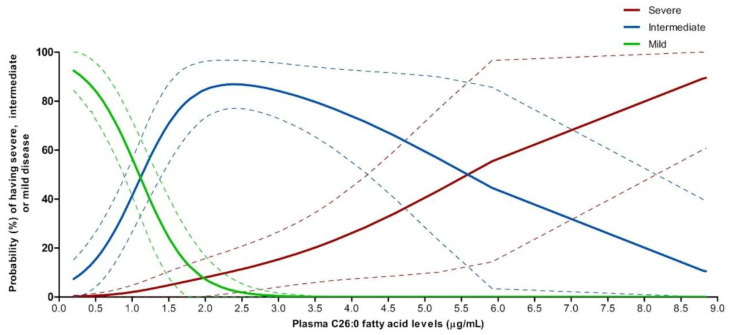
Predicted probabilities of each severity category by plasma C26:0 fatty acid levels at any age. Univariable multinomial logistic model was used to predict the probabilities of having severe (red), intermediate (blue), or mild (green) overall disease severity from plasma C26:0 fatty acid levels measured in 8 severe, 44 intermediate, and 36 mild ZSD individuals (total 88 patients) from our natural history study. The 95% confidence intervals are shown as the dotted lines.

**Table 1 cells-11-01891-t001:** Prevalence of clinical findings by severity in cohort studies.

	Severe (9 Studies Total)		Intermediate (10 Studies Total)		Mild (10 Studies Total)	
Clinical Finding	Studies	% Subjects with Clinical Finding	Studies	% Subjects with Clinical Finding	Studies	% Subjects with Clinical Finding
**Neurological findings**						
Seizure disorder	9	70.8 (51/72)	6	51.0 (26/51) *	5	15.9 (10/63) ^^
Abnormal EEG	5	51.2 (21/41)	1	100.0 (2/2)	1	0 (0/3)
Brain abnormalities	6	69.4 (34/49)	4	15.4 (4/26) **	3	70.6 (24/34) ^^
Ataxia	0	n/a	2	66.7 (10/15)	6	64.3 (18/39)
Hypotonia	5	94.6 (35/37)	7	97.6 (40/41)	4	50 (21/42) ^^
**Gastrointestinal findings**						
Feeding difficulties	5	71.1 (32/45)	3	52.9 (9/17)	3	14.8 (8/54) ^
Gastroesophageal reflux	0	n/a	1	63.6 (7/11)	1	14.3 (1/7)
Abnormal liver function/structure	8	81.5 (53/65)	7	79.0 (30/38)	5	54.3 (38/70) ^^
**Other clinical findings**						
Vision loss	6	58.5 (31/53)	6	92.1 (35/38) **	7	92.7 (63/68)
Hearing loss	2	36.8 (7/19)	5	77.5 (31/40) *	7	52.0 (39/75) ^
Renal cortical microcysts	5	41.9 (18/43)	1	0 (0/24) **	0	n/a
Adrenal insufficiency	1	7.1 (1/14)	3	57.1 (16/28) *	3	11.8 (4/34) ^^
**Developmental outcomes**						
Global developmental delay	3	76.9 (30/39)	6	89.2 (33/37)	5	84.6 (33/39)
Reduced verbal communication	0	n/a	3	63.2 (12/19)	4	36.9 (24/65)
Reduced mobility	0	n/a	5	70.8 (17/24)	4	35.7 (25/70) ^
**Other**						
Shortened lifespan (died ≤ 2 years)	7	83.3 (45/54)	3	24.4 (11/45) **	0	0 (0/77) ^^

Cohort studies include studies that included 3 or more patients with ZSD. There were 5 cohort studies that included only severe ZSD patients, 6 studies that included only intermediate ZSD patients, and 6 studies that included only mild ZSD patients. Two studies included patients in the severe and intermediate category, and one study included patients in the intermediate and mild category. Two studies include patients in all three severity categorizations. Brain abnormalities include those observed by MRI or upon autopsy. Values with an asterisk (*) or double asterisk (**) are significantly different from the severe category for the same outcome variable (*p* < 0.025 or *p* < 0.001, respectively). Values with a caret (^) or double caret (^^) are significantly different from the intermediate category for the same outcome variable (*p* < 0.025 or *p* < 0.001, respectively).

**Table 2 cells-11-01891-t002:** Prevalence of clinical findings by severity in case studies.

	Severe (38 Studies)	Intermediate (18 Studies)	Mild (31 Studies)
Clinical Finding	% Subjects with Clinical Finding	% Subjects with Clinical Finding	% Subjects with Clinical Finding
**Neurological findings**			
Seizure disorder	97.0 (32/33)	83.3 (10/12)	50.0 (4/8)
Abnormal EEG	81.3 (13/16)	72.7 (8/11)	80.0 (4/5)
Brain abnormalities	85.3 (29/34)	76.9 (10/13)	66.7 (12/18)
Ataxia	100.0 (1/1)	100.0 (1/1)	83.3 (5/6)
Hypotonia	100.0 (42/42)	100.0 (17/17)	85.7 (12/14)
**Gastrointestinal findings**			
Feeding difficulties	95.8 (23/24)	100.0 (5/5)	90.0 (9/10)
Abnormal liver function/structure	95.1 (39/41)	93.8 (15/16)	77.8 (14/18)
**Other clinical findings**			
Renal cortical microcysts	90.5 (19/21)	0 (0/2) *	0 (0/5)
Adrenal insufficiency	100.0 (4/4)	62.5 (5/8)	66.7 (2/3)
**Developmental outcomes**			
Global developmental delay	100.0 (9/9)	100.0 (17/17)	90.5 (19/21)
Reduced verbal communication	100.0 (1/1)	100.0 (6/6)	77.8 (14/18)
Reduced mobility	100.0 (1/1)	83.3 (5/6)	86.7 (13/15)
**Other**			
Shortened lifespan (died ≤ 2 years)	93.0(40/43)	31.3 (5/16) **	5.7 (2/35) ^

Case studies were identified as a study that described 2 or fewer subjects. There were 36 case studies studied patients in the severe category, 16 that studied patients in the intermediate category and 31 that reported on patients in the mild category of severity. Two case studies included one severe and one intermediate patient. Brain abnormalities include those observed by MRI or upon autopsy. Values with an asterisk (*) or double asterisk (**) are significantly different from the severe category for the same outcome variable (*p* < 0.025 or *p* < 0.001, respectively). Values with a caret (^) are significantly different from the intermediate category for the same outcome variable (*p* < 0.025).

**Table 3 cells-11-01891-t003:** Prevalence and age of onset of clinical findings in ZSD patients from the natural history study.

	% of Patients with Clinical Finding (Total *n*)				Median Age of Onset * (Q1–Q3) (y)		
Clinical Findings	Severe	Intermediate	Mild	Total All Severities	Severe	Intermediate	Mild
**Neurological findings**							
Seizure disorder	100 (23)	41.3 (63) **	16.3 (49) ^	42.2 (135)	0 (0–0.2)	3.3 (1.0–8.3) **	6.2 (4.0–21.3) ^^
Abnormal EEG	100 (17)	72.2 (36) *	40.0 (10)	74.6 (63)	0 (0–0.1)	2.3 (0.9–5.3) **	5.8 (2.3–36.6) ^^
Brain MRI abnormalities	95.0 (20)	81.0 (42)	75.0 (36)	81.6 (98)	0 (0–0.1)	2.0 (0.4–3.8) **	6 (2.8–19.9) ^^
Hypotonia	100 (23)	98.2 (56)	72.1 (43) ^	89.3 (122)	0.1 (0.1–0.3)	1.5 (0.4–3.6) **	4.9 (1.6–8.2) ^^
**Gastrointestinal findings**							
Feeding difficulties	90.0 (20)	71.9 (57)	0 (48) ^^	47.2 (125)	0.1 (0–0.1)	2.2 (1.3–4.7) **	n/a
Failure to thrive	100 (1)	94.1 (17)	5.3 (19) ^^	48.7 (37)		0.2 (0–0.9)	0.5
Gastroesophageal reflux	28.6 (21)	51.7 (58)	11.1 (45) ^^	33.1 (124)	0.3 (0.3–0.6)	2.4 (1.0–10.6) *	6.5 (2.0–9.2) ^
Abnormal liver functions	94.4 (18)	92.9 (56)	65.9 (44) ^^	83.1 (118)	0.1 (0–0.2)	1.3 (0.3–4.2) **	3.1 (1.3–6.1) ^^
Abnormal liver structure	33.3 (21)	61.4 (57)	31.0 (42) ^	45.8 (120)	0.1 (0.1–0.5)	1.3 (0.6–2.0) **	2.9 (1.1–5.1) ^^
**Other clinical findings**							
Vision loss	100 (9)	89.1 (55)	67.4 (46) ^	80.9 (110)	0.4 (0.2–0.5)	1.8 (0.8–3.6) **	4.4 (1.9–8.2) ^^
Renal cortical microcysts	79.0 (19)	0 (37) **	0 (29)	17.7 (85)	0 (0–0.1)	n/a	n/a
Adrenal insufficiency	14.3 (21)	54.2 (59) *	10.4 (48) ^^	31.3 (128)	0.1 (0–0.1)	3.8 (2.2–11.0) **	20.7 (16.7–23.8) ^^
Cardiac abnormalities	81.3 (16)	17.7 (34) **	20.0 (25)	32 (75)	0 (0–0.1)	0.1 (0–0.8)	26.4 (24.0–27.0) ^^
Bone fractures	0 (3)	51.2 (41)	27.6 (29)	39.7 (73)	n/a	5.4 (3.0–12.0)	5.4 (4.9–14.0)
**Developmental outcomes**							
Global developmental delay	100 (7)	97.5 (40)	33.3 (36) ^^	69.9 (83)	0.5 (0.2–0.7)	3.3 (1.5–5.0) **	4.2 (2.6–6.8) ^^
Not sitting independently	100 (1)	28.8 (59)	0 (49) ^^	16.5 (109)	1.8	2.8 (2.3–3.9)	n/a
Walking with support	n/a	33.9 (59)	12.2 (49) ^	24.1 (108)	n/a	4 (2.8–7.0)	2.3 (1.6–3.0)
Walking independently	n/a	23.7 (59)	87.8 (49) ^^	52.8 (108)	n/a	2.6 (1.9–3.5)	1.5 (1.3–2.5) ^
No words	100 (5)	71.7 (53)	0 (46) ^^	41.4 (104)	1 (0.5–1.2)	3.5 (2.8–8.3) **	n/a
2–3 words together	0 (5)	5.7 (53)	26.1 (46) ^^	14.4 (104)	n/a	4.1 (2.0–15.5)	4.4 (2.7–6.2)
Full sentences	n/a	1.9 (53)	71.7 (46) ^^	34.3 (99)	n/a	8.3	9.8 (6.0–14.8)
Intellectual disability	n/a	100 (23)	30.0 (30) ^^	62.3 (53)	n/a	7.0 (3.9–18.0)	14.6 (10.5–16.2)
**Peroxisome metabolites**							
Decreased C16:0/C16:0 DMA	100 (4)	83.3 (24)	23.5 (17) ^^	62.2 (45)	0 (0–0.1)	2.6 (0.9–5.3) **	6.7 (4.0–15.8) ^^
Decreased C18:0/C18:0 DMA	100 (4)	91.7 (24)	35.3 (17) ^^	71.1 (45)	0 (0–0.1)	2.6 (0.9–5.3) **	6.7 (4.0–15.8) ^^
**Others**							
Shortened lifespan (died at age ≤ 2 years)	95.7 (23)	0 (64) **	0 (49)	16.2 (136)			

Values with an asterisk (*) or double asterisk (**) are significantly different from the severe category for the same outcome variable (*p* < 0.025 or *p* < 0.001, respectively). Values with a caret (^) or double caret (^^) are significantly different from the intermediate category for the same outcome variable (*p* < 0.025 or *p* < 0.001, respectively). C16 saturated dimethyl acetyl to C16 saturated fatty acid (C16:0 DMA/C16:0) and C18:0 DMA/C18:0 plasmalogen ratios are from red blood cell membranes. * Earliest age at which the clinical findings were reported in available medical charts. n/a: not applicable; SD: standard deviation; Q1: first quartile; Q3: third quartile.

**Table 4 cells-11-01891-t004:** Levels of peroxisome metabolites in ZSD patients from the natural history study.

	Reference Range [41,97]	Severe	Intermediate	Mild
**Median plasma C26:0 fatty acid levels,** Q1–Q3 (μg/mL)	0.14–0.31	3.9 ** 2.7–5.6 (*n* = 8)	2.0 ** 1.3–2.8 (*n* = 44)	0.6 ** 0.4–0.9 (*n* = 36)
**Median serum DHCA levels,** Q1–Q3 (μmol/L)	0–0.1	3.1 * (*n* = 1)	3.2 * 0.5–9.7 (*n* = 17)	0.0 * 0.0–0.8 (*n* = 13)
**Median serum THCA levels,** Q1–Q3 (μmol/L)	0–1.3	22.5 * (*n* = 1)	0.6 * 0.2–1.5 (*n* = 17)	0.2 * 0.2–0.4 (*n* = 14)
**Median RBC membrane C16:0 DMA/C16:0 ratio,** Q1–Q3	0.08–0.13	0.007 ** 0.005–0.01 (*n* = 4)	0.056 ** 0.04–0.07 (*n* = 24)	0.094 ** 0.087–0.108 (*n* = 17)
**Median RBC membrane C18:0 DMA/C18:0 ratio,** Q1–Q3	0.20–0.28	0.006 ** 0.004–0.008 (*n* = 4)	0.122 ** 0.1–0.16 (*n* = 24)	0.215 ** 0.182–0.23 (*n* = 17)

Values with an asterisk (*) or double asterisk (**) are significantly different across all three severity categories for the same peroxisome metabolite (*p* < 0.05 or *p* < 0.001, respectively). Q1: first quartile; Q3: third quartile; DHCA: dihydroxycholestanoic acid; THCA: trihydroxycholestanoic acid; RBC: red blood cell; DMA: dimethyl acetyl.

## Data Availability

All data obtained and analyzed in this study are included in this article and in the Appendix A. Raw data supporting the findings of this study are available from the corresponding authors (M.B. and C.Y.) upon request.

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
