# Peer review of "Characterization of Severity in Zellweger Spectrum Disorder by Clinical Findings: A Scoping Review, Meta-Analysis and Medical Chart Review"

_cells, 2022, doi:10.3390/cells11121891_

Round 1
Reviewer 1 Report
The manuscript by Bose et al is a meta analysis/review about Zellweger syndrome preveilance. Furthermore, the authors found the correlation between plasma LCVFA and severity of the disease. In principle the manuscript is scientifically sound and interesting, yet in my opinion it lacks the molecular part behind presented studies. Therefore, I suggest few changes to improve the manuscript:
- The molecular basis of ZS should be provided. Why ZS develops? Which genes are mutated? What are the consequences of these mutations for peroxisome biogenesis/homeostasis?
- Why elevated LCVFA appear in ZS should be clearly stated. Can authors try to explain the correlation between LCVFA level, ZS severity and the molecular alterations that stand behind particular ZS case? (e.g. frequency of particular PEX mutation vs LCFA vs ZS severity.
Author Response
Reviewer 1
The manuscript by Bose et al is a meta-analysis/review about Zellweger syndrome prevalence. Furthermore, the authors found the correlation between plasma LCVFA and severity of the disease. In principle the manuscript is scientifically sound and interesting, yet in my opinion it lacks the molecular part behind presented studies. Therefore, I suggest few changes to improve the manuscript:
- The molecular basis of ZS should be provided. Why ZS develops? Which genes are mutated? What are the consequences of these mutations for peroxisome biogenesis/homeostasis?
We thank the reviewer for this comment. We have expanded upon our Introduction section to include information about why ZSD develops and what the consequences of mutations are with respect to peroxisome biogenesis and function.
- Why elevated LCVFA appear in ZS should be clearly stated. Can authors try to explain the correlation between LCVFA level, ZS severity and the molecular alterations that stand behind particular ZS case? (e.g. frequency of particular PEX mutation vs LCFA vs ZS severity.
This point is also addressed in the Introduction section.
Reviewer 2 Report
In this review the clinical data of ZSD patients that were either reported in cohort studies or in case studies, and in addition from the medical records of patients collected in McGill were brought together. Important strengths are the large number of patients, which allowed to stratify the patients in severe, medium and mild disease based on distinguishing features. Also, the authors critically discussed the limitations of their study. This overview is the first of its kind and will be extremely valuable for clinicians dealing with ZSD patients.
Minor comments
Line 46:’ ZSD are caused by mutations in 14 PEX genes’. I suppose that PEX11β is included, but it would be better to define that this peroxin is involved in peroxisome proliferation/fission and not in the formation of the peroxisomal membrane or in matrix protein import
Is there any overlap between the patients of the cohort/case studies and those of the natural history?
It has been reported before that metabolite levels can change (often diminish) with the age of the ZSD patients. Here, it is not clear at which age the presented data were taken. Why is not the ratio of C26/C22 presented and the ratio of immature C27 bile acids/mature C24 bile acids? At least the potential impact of age on the metabolite levels, and how this may affect the interpretation of the data should be discussed.
Brain abnormalities in Table 1 and 2 : add in the legend what this refers to
Some typos need to be corrected in the following lines (87, 117, 188, 336, 456, 542)
Author Response
Reviewer 2
- In this review the clinical data of ZSD patients that were either reported in cohort studies or in case studies, and in addition from the medical records of patients collected in McGill were brought together. Important strengths are the large number of patients, which allowed to stratify the patients in severe, medium and mild disease based on distinguishing features. Also, the authors critically discussed the limitations of their study. This overview is the first of its kind and will be extremely valuable for clinicians dealing with ZSD patients.
We thank the reviewer for their comments on the value of this manuscript.
- Minor comments
Line 46:’ ZSD are caused by mutations in 14 PEX genes’. I suppose that PEX11β is included, but it would be better to define that this peroxin is involved in peroxisome proliferation/fission and not in the formation of the peroxisomal membrane or in matrix protein import
We thank the reviewer for this observation and suggested correction. We have amended our broad statement in the Introduction section regarding the functions of peroxins to include their role in peroxisome proliferation and fission.
Is there any overlap between the patients of the cohort/case studies and those of the natural history?
We thank the reviewer for this query. We did exclude multiple papers as there appeared to be considerable redundancy in subjects in another included cohort study or in the natural history study. We have now included a statement in the Results section to reflect that. Given the large amount of published case studies on patients with ZSD, we cannot rule out some overlap across patients in the natural history study. As a result, the total reported sample size of all combined datasets may be overestimated. But we expect this overlap to be minimal, and we have listed this as a limitation of our study in the Discussion section of the manuscript.
- It has been reported before that metabolite levels can change (often diminish) with the age of the ZSD patients. Here, it is not clear at which age the presented data were taken. Why is not the ratio of C26/C22 presented and the ratio of immature C27 bile acids/mature C24 bile acids? At least the potential impact of age on the metabolite levels, and how this may affect the interpretation of the data should be discussed.
We thank the reviewer for this significant point regarding the age of subjects relating to changes in metabolite levels. While we did document age wherever possible in our data organization, we did not stratify clinical nor biochemical findings by age in our analysis, as this categorization would result in sample sizes too low to reach significance across severity levels. We have listed this as a limitation and have cited the important study by Wangler, et al. (2018) highlighting the impact of age on metabolite levels in ZSD in our Discussion section. We recognize the reviewer’s point regarding including the ratio of plasma C26:0/C22:0 fatty acid levels, however, we expect that analysis of plasma C26:0/C22:0 fatty acid ratio compared to absolute plasma C26:0 fatty acid values would result in similar findings. We chose to include absolute plasma C26:0 fatty acid values and use this parameter for our model because we expect that the measurement of plasma C26:0 fatty acid levels alone would yield greater ease of use of our model in clinical settings. We did not include plasma C24 bile acid levels as this information was reported less frequently in the natural history data compared to the plasma C27 bile acid levels.
Reference cited:
Wangler MF, Hubert L, Donti TR, Ventura MJ, Miller MJ, Braverman N, Gawron K, Bose M, Moser AB, Jones RO, Rizzo WB, Sutton VR, Sun Q, Kennedy AD, Elsea SH. A metabolomic map of Zellweger spectrum disorders reveals novel disease biomarkers. Genet Med. 2018 Oct;20(10):1274-1283. doi: 10.1038/gim.2017.262.
- Brain abnormalities in Table 1 and 2 : add in the legend what this refers to
We have added a statement in the descriptions for Tables 1 and 2 that describes brain abnormalities as those observed by MRI or by autopsy. Specific observations that were documented as a brain abnormality are described in detail in the Methods Section.
- Some typos need to be corrected in the following lines (87, 117, 188, 336, 456, 542)
We thank the reviewer for directing us to these errors. These errors have been corrected.
Reviewer 3 Report
Excellent compilation of all the clinical signs of Zellweger syndrome. The most important limitation of this kind of study is the retrospective analysis of clinical description. All the patients are not described with the same criteria and therefore it's difficult to have very strong conclusions.
Author Response
Reviewer 3
- Excellent compilation of all the clinical signs of Zellweger syndrome. The most important limitation of this kind of study is the retrospective analysis of clinical description. All the patients are not described with the same criteria and therefore it's difficult to have very strong conclusions.
We thank the reviewer for this favorable comment. We have included an additional statement in the Discussion section about how the retrospective nature of our data evaluation limits our ability to apply the same criteria to specific clinical findings.
Reviewer 4 Report
The manuscript entitled " Characterization of Severity in Zellweger Spectrum Disorder by Clinical Findings: A Scoping Review, Meta-analysis and Medical Chart Review " is well written, has important clinical message, and should be of great interest to the readers, and not only those interested by peroxisomal disorders. This study, submitted to the journal Cells propose a review and meta-analysis of clinical findings and survival in Zellweger spectrum disorder (ZSD) in published 107 studies (including 307 patients), in addition to 136 ZSD individuals from the authors ‘studies. The methodology and the results are sounding, highly interesting and would certainly make this article as a reference study in the field.
Introduction :
Methods :
Did the authors used PubMed as it or MeSH database search? Please add the exact website link.
Results:
I do not understand the choice of some percentage calculation : for example, in Table 3, for the % of patients with clinical finding, the total n = 135, however, the reported % is relative to each category (Severe, Intermediate or Mild). The clinical findings are the same in each category, so why authors didn’t calculate the % to total n?
Particularly in Table 3, clinical symptoms relative to organ functions are presented in the same table as walking sitting, intellectual disabilities or plasmalogen levels. It will be clearer and digest to the readers (particularly non clinicians) to put them in separate tables.
Regarding “ Predicted probabilities of disease severity by plasma C26:0 VLCFA levels”, authors reported that “ the model showed that C26:0 fatty acid levels of 1.08 microg/mL corresponded to equal chance of having an intermediate or a mild disease, while levels of 5.18 microg/mL corresponded to equal chance of having an intermediate or a severe disease. “ Regarding the reference range, we can see that from intermediate (3 folds to reference range) to mild disease (9 folds to reference range), the multiplier for VLCFA level is 3, while from mild (9 folds to reference range) to severe disease (18 folds to reference range), this multiplier is 2. Could we talk about more a risk factor regarding the normal subjects and/or on an equal chance regarding the severity of disease stage ? authors add comments on this?
Author Response
Reviewer 4
The manuscript entitled " Characterization of Severity in Zellweger Spectrum Disorder by Clinical Findings: A Scoping Review, Meta-analysis and Medical Chart Review " is well written, has important clinical message, and should be of great interest to the readers, and not only those interested by peroxisomal disorders. This study, submitted to the journal Cells propose a review and meta-analysis of clinical findings and survival in Zellweger spectrum disorder (ZSD) in published 107 studies (including 307 patients), in addition to 136 ZSD individuals from the authors ‘studies. The methodology and the results are sounding, highly interesting and would certainly make this article as a reference study in the field.
We thank the reviewer for this comment on the potential value and broad interest of our manuscript.
Introduction :
Methods :
- Did the authors used PubMed as it or MeSH database search? Please add the exact website link.
The website for PubMed has been included in the manuscript.
Results:
- I do not understand the choice of some percentage calculation : for example, in Table 3, for the % of patients with clinical finding, the total n = 135, however, the reported % is relative to each category (Severe, Intermediate or Mild). The clinical findings are the same in each category, so why authors didn’t calculate the % to total n?
We thank the reviewer for this comment. We have included a column in Table 3 for the percentage of all subjects that present with the corresponding clinical finding.
- Particularly in Table 3, clinical symptoms relative to organ functions are presented in the same table as walking sitting, intellectual disabilities or plasmalogen levels. It will be clearer and digest to the readers (particularly non clinicians) to put them in separate tables.
We thank the reviewer for this comment and recognize the value of categorizing these clinical findings for clarity. Since we already have seven tables and figures for this manuscript, as well as supplementary tables and figures, instead of separating data into multiple tables, we grouped each of the clinical and biochemical findings for Tables 1-3 into “Neurological findings, Gastrointestinal findings, Other clinical findings, Developmental outcomes, Peroxisome metabolites and Others” to improve the readability of the tables.
- Regarding “ Predicted probabilities of disease severity by plasma C26:0 VLCFA levels”, authors reported that “ the model showed that C26:0 fatty acid levels of 1.08 microg/mL corresponded to equal chance of having an intermediate or a mild disease, while levels of 5.18 microg/mL corresponded to equal chance of having an intermediate or a severe disease. “ Regarding the reference range, we can see that from intermediate (3 folds to reference range) to mild disease (9 folds to reference range), the multiplier for VLCFA level is 3, while from mild (9 folds to reference range) to severe disease (18 folds to reference range), this multiplier is 2. Could we talk about more a risk factor regarding the normal subjects and/or on an equal chance regarding the severity of disease stage ? authors add comments on this?
We thank the reviewer for this comment. We have included statements in the Discussion section that present the data with respect to the reference range and as well as the limitations of the model within specific ranges of plasma C26:0 levels where probabilities of severity designations are similar in multiple categories.